# Presence of Zonula Occludens Toxin-Coding Genes among *Vibrio parahaemolyticus* Isolates of Clinical and Environmental Origin

**DOI:** 10.3390/microorganisms12030504

**Published:** 2024-02-29

**Authors:** Cristian Iribarren, Nicolás Plaza, Sebastián Ramírez-Araya, Diliana Pérez-Reytor, Ítalo M. Urrutia, Elisabetta Suffredini, Teresa Vicenza, Soledad Ulloa, Jorge Fernández, Paola Navarrete, Victor Jaña, Leonardo Pavez, Talía del Pozo, Gino Corsini, Carmen Lopez-Joven, Katherine García

**Affiliations:** 1Instituto de Ciencias Biomédicas, Facultad de Ciencias de la Salud, Universidad Autónoma de Chile, Santiago 8320000, Chile; cristian.iribarren@uautonoma.cl (C.I.); nicolas.plaza@uautonoma.cl (N.P.); seba264@gmail.com (S.R.-A.); d.perez@uautonoma.cl (D.P.-R.); italo.urrutia@uautonoma.cl (Í.M.U.); gino.corsini@uautonoma.cl (G.C.); 2Department of Food Safety, Nutrition and Veterinary Public Health, Istituto Superiore di Sanità, 00161 Rome, Italy; elisabetta.suffredini@iss.it (E.S.); teresa.vicenza@iss.it (T.V.); 3Subdepartment of Genomic and Molecular Genetics, Public Health Institute of Chile, Santiago 7780050, Chile; sole.ulloa@gmail.com (S.U.); jfernand@ispch.cl (J.F.); 4Microbiology and Probiotics Laboratory, Institute of Nutrition and Food Technology (INTA), University of Chile, Santiago 7830490, Chile; pnavarre@inta.uchile.cl; 5Núcleo de Investigaciones Aplicadas en Ciencias Veterinarias y Agronómicas (NIAVA), Universidad de Las Américas, Santiago 7500975, Chile; victor.jgaray@gmail.com (V.J.); tdelpozo@udla.cl (T.d.P.); 6Núcleo de Investigaciones en Ciencias Biológicas, Facultad de Medicina Veterinaria y Agronomía, Universidad de Las Américas, Santiago 7500000, Chile; lpavez@udla.cl; 7Instituto de Medicina Preventiva Veterinaria, Facultad de Ciencias Veterinarias, Universidad Austral de Chile, Valdivia 5090000, Chile

**Keywords:** *Vibrio parahaemolyticus*, *Zonula occludens*, toxin, *zot*, secretion systems, bivalves

## Abstract

In recent studies, emphasis has been placed on the zonula occludens toxin (Zot) from the non-toxigenic *Vibrio parahaemolyticus* strain PMC53.7 as an agent inducing alterations in the actin cytoskeleton of infected Caco-2 cells and which appears as a relevant virulence factor. Universal *zot* primers were designed by the alignment of different types of *zot* gene and identification of conserved sequences to investigate the presence in diverse environmental and clinical *V. parahaemolyticus* isolates, in co-occurrence with virulence factors, such a hemolysins and secretion systems. The study screened a total of 390 isolates from environmental sources from Chile and Italy and 95 Chilean clinical isolates. The results revealed that around 37.2% of Chilean environmental strains and 25.9% of Italian strains, and 24.2% of clinical isolates carried the *zot* gene. The Zot-C2 cluster was present in 71.4% of Chilean environmental strains but absent in clinical isolates, while the Zot-C4 cluster was identified in 28.6% of environmental and 100% of clinical isolates. Understanding the role of *zot* in *V. parahaemolyticus* virulence is crucial, especially considering the risk associated with consuming diverse isolates from bivalves and the co-occurrence with virulence factors such as TDH, TRH or T3SS2.

## 1. Introduction

*Vibrio parahaemolyticus* stands out as an important cause of bacterial gastroenteritis associated with raw seafood consumption at a global level [1,2,3]. The traditional pathogenicity of *V. parahaemolyticus* strains is associated with toxins, namely thermostable direct hemolysin (TDH) and TDH-related hemolysin (TRH), which are encoded by the *tdh* and *trh* genes, respectively. Additionally, it encodes virulence regulatory factors such as ToxR, which is capable of regulating protein nanomachines called secretion systems, specifically Type 3 (T3SS) and Type 6 (T6SS) [4,5]. The T3SS encoded on chromosome 2 (T3SS2), is essential for colonizing the intestine and pathogenicity of pandemic strains [6,7]. All of these are considered the main virulence factors of the pandemic strain. Notably, approximately 10% of *V. parahaemolyticus* clinical isolates from the stools of gastroenteritis patients lack *tdh*, *trh* and T3SS2 (and hence are reported as non-toxigenic), despite having been isolated from stools of gastroenteritis patients [8]. Interestingly, in vitro studies have shown that these non-toxigenic strains of *V. parahaemolyticus* can exhibit significant cytotoxicity toward human gastrointestinal cells [9,10]. Previous research by Lynch and colleagues demonstrated that disruption of the barrier function and cytoskeletal structure can occur independently of TDH and TRH production, as observed in non-toxigenic clinical isolates [11]. More recently, the impact on F-actin organization in Caco-2 cells generated by environmental *V. parahaemolyticus* has also been highlighted [12]. In our previous work, we identified a prophage encoding the zonula occludens toxin (Zot) in the genome of the non-toxigenic and highly cytotoxic clinical strain PMC53.7 (*tdh*-/*trh*-/T3SS2-), which induces significant cytoskeletal alterations in Caco-2 cells [8,13]. The Zot-PMC53.7 sequence conserves motifs in its N-terminal end and shares structural similarities with Zot from *Vibrio cholerae* [13]. It is relevant to consider that in *V. cholerae*, Zot is the second major toxin encoded by a filamentous phage CTXϕ [8,14]. This protein appears to have a dual role, as it plays a critical role in phage CTXϕ morphogenesis, and it has also been documented to exhibit enterotoxicity activity [15]. This contributes to the induction of mild to moderate diarrhea [16] and leads to rearrangement of the actin cytoskeleton and disturbance of paracellular permeability [17]. Prevalence studies of various virulence genes in non-O1/non-O139 *V. cholerae* isolates from California revealed that 17% of the strains carried the *cholera toxin gene* (ctxA), with about one-third of them possessing the *zot* gene [18]. Recent molecular analysis of *V. cholerae* isolates stemming from outbreaks in southern Ghana revealed that 130 of 168 isolates tested positive for *zot* [19]. While there are currently no reported associations of the *zot* gene with diarrhea outbreaks caused by *V. parahaemolyticus*, the absence of such associations may explain the lack of prevalence studies on *zot* in this species. However, the presence of *zot* in non-toxigenic strains of *V. cholerae* raises intriguing questions about its potential role in other bacterial species. Specifically, it raises the necessity to investigate whether ***zot*** might be associated with *V. parahaemolyticus* strains lacking classical virulence markers such as hemolysins or T3SS2, especially clinical isolates. Based on this evidence, we evaluated the presence of *zot* in environmental and clinical isolates of *V. parahaemolyticus* from diverse origins and co-occurrence with *tdh* and *trh* genes. This was performed by designing universal primers targeting conserved regions, that allowed the detection of different groups of Zot and were implemented to detect the presence of *zot* in samples isolated in Chile and the Adriatic Sea. Additionally, we searched for *zot* genes among sequences of genomes obtained from Chilean clinical isolates, to clarify the relevance of a *zot* sequence in isolates of *V. parahaemolyticus* associated with gastroenteritis.

## 2. Materials and Methods

### 2.1. Recollection of Seafood from Market Points and Harvest Areas in Chile

*V. parahaemolyticus* environmental isolates were sourced from *Mytilus chilensis* collected at street markets in Santiago Metropolitan Region between 2018 and 2020 and vested within the Valdivia River estuary in Corral Bay, Los Rios Region and Reloncaví Fjord in the Los Lagos Region from 2017 to 2018 along of the southeastern Pacific coast of Chile. 

### 2.2. Environmental Strains Isolation

In the Metropolitan Region, a total of three pooled samples per street market (twelve locations) were processed (ten individual mussels per pool sample). Instead, in the Valdivia River estuary and Reloncaví Fjord a total of five pooled samples per area and month were randomly collected (five individuals of mussels per pool sample) [20]. Mussels were scrubbed with a brush and washed under running tap water to remove any shell mud or dirt. Subsequently, using sterile materials, they were opened to retrieve the soft tissue of the shellfish. The collected tissue was homogenized and enriched for *V. parahaemolyticus* in alkaline peptone water medium for 24 h at 37 °C. Then, one aliquot from each homogenized pool (100 µL) or enrichment tube (20 µL) was plated on selective media CHROMagar *Vibrio* plates (CHROMagar Microbiology, Paris, France), and incubated at 37 °C for 24 h.

### 2.3. V. parahaemolyticus Identification

Bacterial colonies with the morphology and mauve color expected for *V. parahaemolyticus* were selected and purified for DNA extraction following the manufacturer’s protocol (Wizard Genomic DNA Kit, Promega, Madison, WI, USA). The DNA concentration was adjusted to 40–50 ng/μL, and PCR analysis was performed using 10 ng of total bacterial DNA per reaction. The previously described *V. parahaemolyticus* primers targeting *tlh*, *toxR*, *tdh* and *trh* genes were employed for identification and characterization of pathogenicity profile [21,22,23].

### 2.4. Phylogeny Analyses of Zot Sequence in Database

To determine the divergence of Zot nucleotide sequences of *V. parahaemolyticus* available in NCBI (August 2019), for the design of universal primers, we constructed a phylogenetic tree of the *zot* family. We generated an alignment using 22 Zot nucleotide sequences from *V. parahaemolyticus* and one Zot nucleotide sequence from *V. cholerae* (outgroup), performed with MAFFT using L-INS-i iterative refinement [24,25,26] and IQ-TREE [27,28,29] for prediction of the best-suited model and phylogenetic construction with standard parameters. The phylogenetic tree was visualized with the ggtree v3.8.2 package in R [30] and grouped according to sequence identity. 

### 2.5. Primers Design, Amplification and Sequencing

Conserved regions were identified with Bioedit Sequence Alignment Editor software (version 7.0.5.3) using a search criterion of a minimum eight consecutive nucleotides between the 22 sequences. They were subsequently used to manually design specific primers, named Zot-Vp Forward and Reverse: Zot-VpF: 5′-TCCCTGCACTTCGTGAAG-3′ and Zot-VpR: 5′-TTCGCCCGTTTGAAACCTGC-3′, which were analyzed and validated using OligoAnalyzer and Primer Blast [31,32]. The PCR mix included 10 uL of 2× OneTaq Master Mix (New England Biolabs, Ipswich, MA, USA), 1 uL of each primer to 10 pmol/uL (forward and reverse), 6 uL of free nuclease water and 2 uL of genomic DNA 40–50 ng/μL. To standardize Zot-Vp primers, we purified DNA from clinical strains PMC53.7, VpKX (RIMD2210633) and *V. cholerae* N16961 strain. The thermal profile used was as follows: 1 cycle of 94 °C for 5 min, 35 cycles of denaturing (94 °C/30 s), annealing (59 °C/15 s) and extension (72 °C/20 s) and one final cycle of 72 °C for 10 min. PCR amplicons were analyzed by electrophoresis in 1× agarose gel and purified using the GeneJET PCR Purification Kit (Thermo Fisher Scientific, Waltham, MA, USA). Fragments were sequenced on both strands by Macrogen (Seoul, Republic of Korea), and the resulting consensus sequences were then analyzed using BLAST to confirm their identity. 

### 2.6. Presence of Zot in Isolates from the Adriatic Sea (Italy)

The 54 Italian environmental *V. parahaemolyticus* isolates from bivalve shellfish (*Mytilus galloprovincialis* and *Tapes philippinarum*) harvested in the Northern Adriatic Sea between 2012 and 2016 were obtained following the ISO/TS 21872-1:2007 procedure [33]. TCBS and CHROMagar *Vibrio* were used as the selective media to isolate the colonies. The DNA extraction for the Italian isolates were purified using the GRS Genomic DNA Kit (Grisp, Porto, Portugal). Primers targeting the *toxR*, *tdh*, *trh*, T3SS1 and T3SS2 genes [21,22,23] were employed for strain identification and the universal primers Zot-Vp were tested. PCR amplicons were purified using the GRS PCR&Gel band Purification kit (Grisp, Porto, Portugal) and sequenced on both strands by Eurofins Genomics (Ebersberg, Germany). Consensus sequences were provided for analysis as described for Chilean *V. parahaemolyticus* environmental isolates.

### 2.7. Genome Analysis of Clinical Isolates

We analyzed the presence of the *zot* gene on the 95 Chilean clinical isolates provided by the Institute of Public Health of Chile (ISP). Sequences of clinical *V. parahaemolyticus* isolates from human stools were examined from the ISP database, which contains information of strains between 2012 and 2018, using the Basic Local Alignment Search Tool (BLAST) and the reference strains used were NC_002473.1 (pO3K6) and CP013248.1 (FORC_022_chr-I).

## 3. Results

### 3.1. Availability of Zot Sequences Described in V. parahaemolyticus

Based on sequence data available in the NCBI until August 2019, we identified a total of 22 sequences described as *zot* or *zot*-related in *V. parahaemolyticus* (Table 1). Among these, 59.1% were obtained from isolates originating in Southeast Asia (including India, Bangladesh, China, Japan and South Korea), while 22.7% were sourced from the United States, and the 13.6% were attributed to South America. These isolates encompassed environmental (59.1%) and clinical samples (40.9%). Notably, two *V. parahaemolyticus* strains isolated in South Korea, FORC_006 and FORC_014, exhibited a distinctive feature: the presence of two copies of the *zot* gene. One copy was encoded on chromosome one (chr-I), while the other was on chromosome two (chr-II) as detailed in Table 1.

### 3.2. Identification of Conserved Regions in V. parahaemolyticus Zot-Sequences and Primers Design

*Zot* sequences were categorized into four distinct clusters by phylogenetic reconstruction (Figure 1A). In particular, the first cluster (Zot-C1) encompassed five sequences originating from South Korea, exhibiting sequence identities ranging from 96% to 100% (Figure 1B). The second cluster (Zot-C2) presented only two sequences, from the Vp_FORC_022 and PMC53.7 strain, sharing a sequence identity of 98% (Figure 1B). The third cluster (Zot-C3) contained *zot* sequence from UCM-V493 and FORC_014 strain, that exhibited a sequence identity of 82% between them (Figure 1B). The fourth and final cluster (Zot-C4) comprised a significant 59% of the *zot* sequences and notably incorporated the pandemic strain RIMD2210663 (VpKX), the sequence identities spanning from 98% to 100% of identity (Figure 1B). Although, the identity withing the Zot-C3 cluster was relatively lower in comparison to the total number of sequences analyzed in this study. It is noteworthy that although the similarity percentage within sequences belonging to each individual cluster exceeded 95%, the similarity percentage across different clusters demonstrated a comparatively lower range, varying between 24% and 43.7%, as represented in Figure 1B.

Due to the distinct divergence observed between the Zot-C1 clade (exclusively comprising Asiatic strains) and other clades, and the low percent of identity within the Zot-C3 clade, the primer design strategy was focused on identifying conserved regions shared between the Zot-C2 and Zot-C4 clusters. This analysis revealed seven conserved regions, each spanning eight to ten nucleotides in length. After scrutinizing thermodynamic properties in silico, the optimal primer pair was designed as: Zot-VpFw: 5′-TCCCTGCACTTCGTGAAG-3′ and Zot-VpRv: 5′-TTCGCCCGTTTGAAACCTGC-3′, with an expected product size of 237 base pairs with a Tm (57.3–59 °C), %GC content (55–57.9) and potential homo/heterodimer formation, based on ΔG values (−5.09 to −1.34 Kcal/mol).

### 3.3. Primers Specificity and Amplicon Analysis 

The specificity of the designed primers were tested with gDNA from the *V. cholerae* strain N16961 (O1, biovar El Tor), which carries a different sequence of *zot* (outgroup in a phylogenetics tree), compared to the non-toxigenic *V. parahaemolyticus* PMC53.7 strain, in which the presence of the *zot* (Zot-C2) has been described [11,16]. The best amplification conditions for PCR were set at 57–59 °C of hybridization temperature, 35 cycles on the PCR program, 50–200 ng of gDNA and 10 to 15 pmol of the amount of each primer (Appendix A). As expected, the Zot-Vp primers did not amplify the zot gene of *V. cholerae* (Figure 2, Lane 4), but this gene was amplified with *zot* primers for *V. cholerae* (Fw_Zot_Vc 5′-CTTCATATGAGTATCTTTATTCATCACGGC-3′ and Rv_Zot_Vc 5′-AACAAGCTTTTAGTGGTGATGATGGTGATGAAATATACTATTTAGTCCTTTTTTATC-3′) [16] (Figure 2, Lane 2). In addition, our evaluation with Zot-Vp primers showed that PMC53.7 produced a band of 237 bp (Figure 2, Lane 5). After sequencing the PCR products, it was confirmed that the PMC53.7 amplified fragment matched with the Zot-C2, aligning with their respective genomic information. 

### 3.4. Presence of Zot Gene in Environmental V. parahaemolyticus Chilean Isolates

To evaluate the prevalence of the *zot* gene, we isolated a total of 336 strains of *V. parahaemolyticus* from the three different locations in Chile (Figure 3A): 59 strains from the Metropolitan Region derived from different street market points scattered across Santiago and encompassed locations such as Estación Central, Independencia, Lo Espejo (Fishing Terminal), Macul, Maipú, Ñuñoa, Pedro Aguirre Cerda (PAC), Peñalolen, Puente Alto, Quinta Normal, San Miguel and Santiago midtown (Figure 3B), 212 strains from Los Rios Region from the subtidal zone at three specified locations within each area: point 1 (39°52′33″ S, 73°23′08″ W), point 2 (39°53′39″ S, 73°23′10″ W) and point 3 (39°52′10″ S, 73°21′19″ W) (Figure 3C) and 65 strains from Los Lagos Region from the intertidal zone at point 4 (41°42′33″ S, 72°36′60″ W), point 6 (41°42′37″ S, 72°36′32″ W) and point 7 (41°42′37″ S, 72°36′06″ W) in Reloncaví Fjord (Figure 3D) [20].

The presence of the *zot* gene in environmental isolates from bivalves obtained from street markets and harvest points was evaluated using the Zot-Vp primers designed, and conditions previously defined for the PCR in this study. First, we detected the presence of the *zot* gene in 47.5% (28/59) of the isolates from street market points (Figure 4A). On the other hand, in harvest points the presence of the *zot* gene was detected in 32.6% [13] (69/212) and 43.1% (28/65) of the isolates from the Los Rios Region and Los Lagos Region, respectively (Figure 4A). The overall presence of the *zot* gene in *V. parahaemolyticus* isolates was therefore estimated at 37.2% (125/336). Furthermore, the sequencing of 35 samples randomly chosen among the 336 amplified ones showed that the Zot-C2 cluster was predominant (25/35 samples, 71.4%) compared to the Zot-C4 cluster (10/35 samples, 28.6%) (Figure 4B).

Additionally, we analyzed the co-occurrence of genes associated with the virulence factors (*tdh* and *trh*) of *V. parahaemolyticus* with the presence of the *zot* gene in each location and isolation year. The *tdh* gene was detected in 23 of the 336 strains analyzed (6.8%), all of them isolated in the Region of Los Rios between the years 2017 and 2018, and only three isolates (0.9%) contained both *zot* and *tdh* (Table 2). Regarding the *trh* gene, it was only detected in 5/336 (1.5%) of the strains (four from the Los Rios Region and one from the Los Lagos Region). In turn, only one isolate (0.3%) from the Los Rios Region contained simultaneously the *zot* and *trh* genes (Table 2).

### 3.5. Presence of Zot Gene in Environmental V. parahaemolyticus Italian Isolates

To validate the universal primers design and to determine if the occurrence of the *zot* gene is inherently present in the *V. parahaemolyticus* species as potential virulence factors, we analyzed 54 environmental isolates of *V. parahaemolyticus* from bivalve shellfish harvested in the Northern Adriatic Sea (Italy). All isolates were negative to *tdh*, while ten isolates displayed the presence of the *trh* gene (Appendix A). Considering *zot*, the gene was amplified in 14 of the 54 isolates (25.9%) (Figure 5A) and only one of the isolates (1221 strain) was positive for the *zot* and *trh* gene (Figure 5B). All zot positive amplicons were sequenced and classified according to the *zot* cluster group, showing the distribution of Zot-C2 (4/14; 28.6%) and Zot-C4 (10/14; 71.4%) clusters (Figure 5C), therefore pointing out the predominance of Zot-C4 compared to Zot-C2 in the Italian isolates, in contrast to the Chilean samples.

### 3.6. Presence of Zot Gene in Clinical V. parahaemolyticus Genomes

Due to the co-occurrence of the *zot* gene with virulence genes such as *tdh* and *trh* (associated with the T3SS of *V. parahaemolyticus*) in Chilean and Italian strains, we decided to analyze samples of clinical origin to evaluate the presence and implication of zot in this type of sample. We analyzed 95 sequences available in the ISP database, identifying 23 of 95 genomes (24.2%) that contained the *zot* gene (Figure 6A). All *zot*-positive isolates were classified into the Zot-C4 cluster group (RIMD2210633 cluster), with the co-occurrence of virulence factor and the *zot* gene in 21 isolates having *tdh* and *trh* genes (22.1%) and 23 isolated with *zot* and *tdh* (24.2%) (Figure 6B).

## 4. Discussion

Several studies have underscored the capacity of *V. parahaemolyticus* environmental strains to exert an impact on cultured human cells, triggering disturbances in both the paracellular pathway and the structural integrity of the cytoskeleton. In previous work, we demonstrated that the zot of PMC53.7, whose sequence clusters within the Zot-C2 group, elicits pronounced alterations in the actin cytoskeletal structure of infected Caco-2 cells [13]. Likewise, the zot protein of RIMD2210633, clustering within Zot-C4, has been identified as an activating guanylate cyclase-enterotoxin through bioinformatics methods [13]. The primary objective of this study was to design universal primers capable of detecting all *zot* sequences present in *V. parahaemolyticus* isolates. However, during this endeavor, we observed that the Zot-C1 cluster group, including sequences exclusively attributed to strains originating from Southeast Asia, formed a distinct and independent group. This geographical separation from strains within other clusters may in part explain the observed genetic divergence [37]. Furthermore, the Zot-C3 cluster group exhibited a notable lower percentage of nucleotide identity in comparison to other clusters. Therefore, our primer design strategy focused on identifying conserved regions shared between the Zot-C2 and Zot-C4 clusters group, both containing pathogenicity Chilean strains (PMC53.7 and RIMD2210633, respectively). Interestingly, our assessment of *zot* within Chilean *V. parahaemolyticus* strains sourced from distinct harvested areas, utilizing primers targeting the conserved sequences among Zot-C2/C4 clusters, revealed a positivity of 32.6% and 43.1% in the Los Rios and Los Lagos Region, respectively. Furthermore, our observations highlighted a *zot* positivity of 47.5% in isolates obtained from street markets. We found that the percentage observed in the presence of *zot* in the Los Lagos region and Metropolitan Region were similar and the presence of *tdh* or co-occurrence (*tdh*-*zot*) in the Los Rios Region is considerably higher than in the other locations. Drawing from these findings, we can infer that the presence of these genes may play a pivotal role in the emergence of nationwide outbreaks. This conjecture is particularly relevant given the Los Lagos Region’s distinction as the main source of these bivalves and the pre-eminent center for mussel extraction in Chile [20]. Notably, the positivity of *zot* within the Los Rios Region was higher than that observed in the Los Lagos Region. Previously, it was observed that the presence of the *tdh* gene is associated with the salinity of the environment [20]. In fact, the *tdh* gene was found in a low percentage in the Los Lagos Region, which has lower salinity than the Los Rios Region. 

In an effort to determine whether the presence of the *zot* gene extends to other locations and to validate the universal primers designed in this study as a new virulence marker for *V. parahaemolyticus*, we tested in isolates from bivalve shellfish harvested in the Northern Adriatic Sea (Italy), the third largest European Union producer of bivalve mollusks. Our analyses showed that 25.9% (14 of 54) of the Italian isolates carried the *zot* gene, which is lower than the values observed in Los Rios (32.6%). Furthermore, this analysis offered further insights into the diverse distribution patterns of *zot*-coding genes in *V. parahaemolyticus*, as the Italian isolates showed a predominance of Zot-C4 sequences (RIMD2210633 group), compared to the Chilean ones, in which the Zot-C2 cluster was preponderant (PMC53.7 group). While the timing and collection strategy of samples between Chilean and Italian strains was not specifically designed for comparison of *zot* occurrence in the two groups, it is noteworthy that the percentage of ***zot***-containing strains in Italy is comparatively lower than what was observed in southern Chilean strains (25.9% *v*/*s* 37.2%, respectively). Future studies are needed to understand whether environmental or anthropogenic factors could be contributing to the variability of the presence of the *zot* gene.

Given the consistent association of *zot* with bacteriophages [8,43] and considering that the N-terminal end of Zot would act in the assembly and transport of phages through the envelope [13,44], it would be relevant to understand how environmental conditions affect the distribution patterns of phages that harbor such toxins. Recent research has suggested that the dispersion of certain vibriophages is contingent upon prevailing environmental conditions, reinforcing the intricate inter-relationship between microbial dynamics and their surroundings [43]. Therefore, it is important to recognize which environmental factors, including salinity, temperature, depth, organic matter and others, may influence the stability of prophages harboring *zot* genes. 

In the analysis of clinical strains, the presence of *zot* within the sequences was determined in 24.2% of the genomes (23 of 95). The emergence of strains that encode the *zot* gene and *tdh* and/or *trh* toxins described in this study is relevant, because this co-occurrence may be the cause of new outbreaks due to infection by *V. parahaemolyticus*. It is well described that both toxins are encoded near to the T3SS2 of *V. parahaemolyticus* [45,46], which is considered a key in the pandemic *V. parahaemolyticus* virulence factor [6,7]. The *tdh* toxin is mainly encoded in the phylotype T3SS2α (RIMD2210633), the *trh* in the T3SS2β (MAVP-R) and in turn, when *tdh* and *trh* are found together, it is the T3SS2γ classification (MAVP-Q) [47]. It should be noted that, in our results, all strains of clinical origin are part of Zot-C4, where the MAVP-26 strain stands out in our analyses (T3SS2γ) corresponding to ST36 [47], which has both toxins (*tdh*/*trh*) and now the presence of the *zot* gene was identified in this strain. This group is the cause of multiple outbreaks in recent years in the northeastern parts of USA and Europe [48,49].

Remarkably, it is worth noting that Zot-C2 was present in 71.4% of Chilean environmental isolates, yet this *zot* variant was conspicuously absent from the clinical isolates (100% Zot-C4). This observation raises intriguing possibilities: it might suggest a heightened stability of strains carrying Zot-C4 types during their transition from the environment to the human gut. Alternatively, this could be explained because cases attributable to strains with the presence of Zot-C2 cause cases that are mild or moderate in severity [17], so the presence of this Zot-C2 cluster could be underestimated in clinical strains that reach the Chilean ISP to be analyzed. Considering these results, it becomes important to investigate the true impact of *zot* belonging to the Zot-C4 cluster in the pathogenicity of *tdh*/*trh* strains of *V. parahaemolyticus*.

## 5. Conclusions

The Los Lagos and Los Rios regions hold significant importance as crucial mussel harvesting zones and the distribution of bivalves intended for human consumption, particularly in the Metropolitan area, is probably influenced by the bivalve supply from the Los Lagos Region. Our study revealed the presence of *zot* in *V. parahaemolyticus* isolates coming from the Los Rios Region (32.6%), Los Lagos Region (43.1%) and the street market of the Metropolitan Region (47.5%). Although we hypothesize that the varying presence between the Los Lagos Region and Los Rios Region could be attributed to salinity variations, other sites must be studied to validate this hypothesis. Our study uncovers diverse presence patterns of *zot*, *tdh* and *trh*-containing strains, which are probably influenced by geographical, environmental and genetic factors. Since the consumption of bivalves implies the intake of different *V. parahaemolyticus* isolates, the intricate relationships between *zot*, bacteriophages, environmental conditions and potential virulence factors warrant further investigation to unravel the true relevance of *zot* in *V. parahaemolyticus* pathogenicity.

## Figures and Tables

**Figure 1 microorganisms-12-00504-f001:**
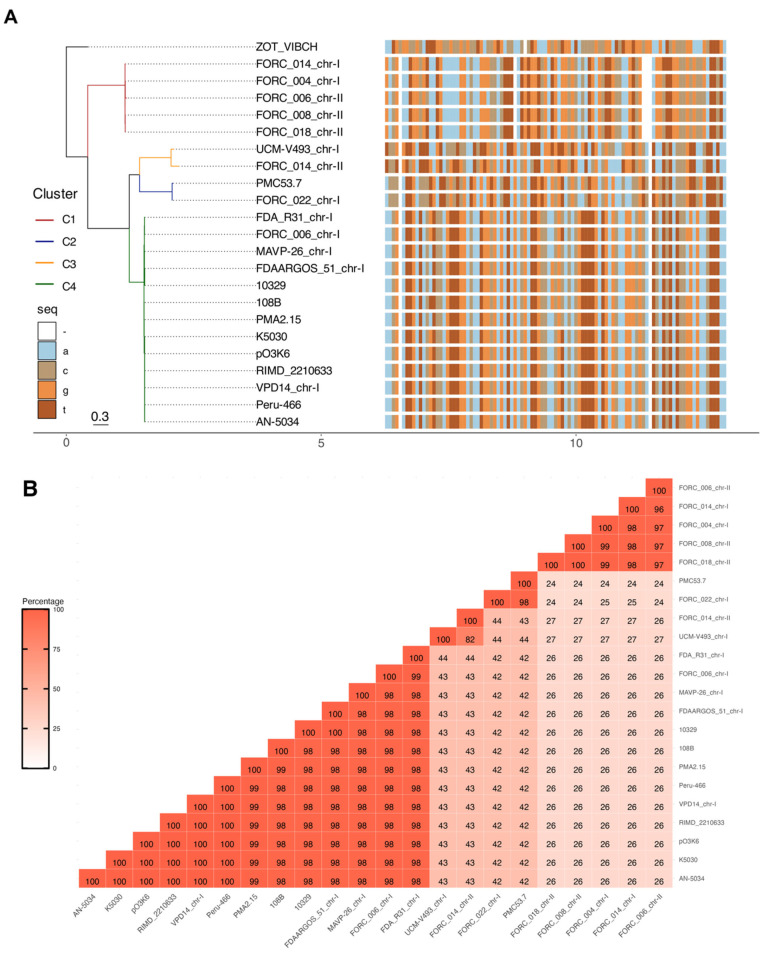
Phylogenetic analysis based on *zot* gene sequences from *V. parahaemolyticus.* (**A**) The tree was constructed after multiple sequence alignment of 22 Zot nucleotide sequences of *V. parahaemolyticus* and one from *V. cholerae* available in NCBI with MAFFT alignment using L-INS-i iterative refinement and IQ-TREE program. The tree shows the bootstrap percentage (from 1000 replicates) next to branches. The sequences have been colored by four different colors shown in the “seq” column for each nucleotide. Color changes on the aligned sequences represent nucleotide differences. All the *zot* sequences were grouped into four clusters. The first cluster (Zot-C1) comprised five sequences from South Korea. The second cluster (Zot-C2) contained PMC53.7-Zot. The third cluster (Zot-C3) has only two sequences that exhibit comparatively lower sequence identity. Finally, the fourth cluster (Zot-C4) comprises the pandemic strain RIMD2210663 (VpKX). (**B**) *zot* sequence percent identity between each strain was calculated based on BLAST and was plotted as color ranges according to the percentages of nucleotide identity.

**Figure 2 microorganisms-12-00504-f002:**
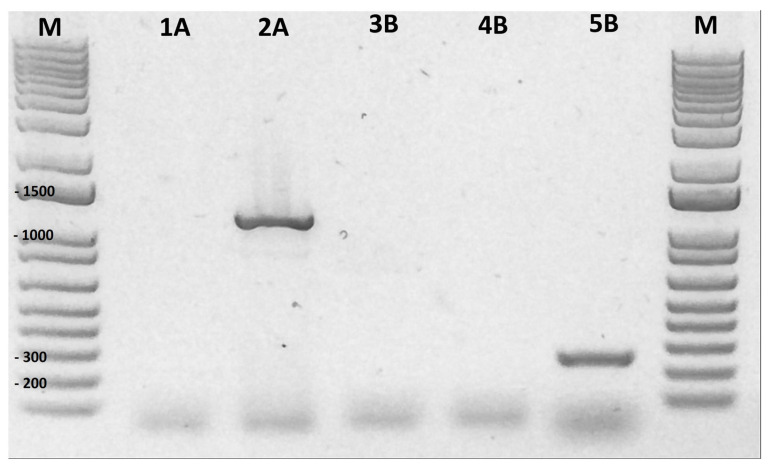
Specificity of *V. parahaemolyticus zot* primers. Assessment of the specificity of *V. parahaemolyticus zot* primers was conducted through the amplification of *zot* from *V. cholerae* and *V. parahaemolyticus*. The gel electrophoresis results are presented in the following manner: Lane M: DNA ladder (1 Kb plus); Lane 1: Negative control without gDNA; Lane 2: gDNA of *V. cholerae* N16961 strain; Lane 3: Negative control without gDNA; Lane 4: gDNA of *V. cholerae* N16961 strain; Lane 5: DNA of *V. parahaemolyticus* PMC53.7 strain. Image of the agarose gel at 1% *w*/*v*. The size of the ladder bands is indicated which allows the identification of the resulting amplicon size. A: *V. cholerae zot* primers. B: *V. parahaemolyticus* Zot-Vp primers elaborated in this study.

**Figure 3 microorganisms-12-00504-f003:**
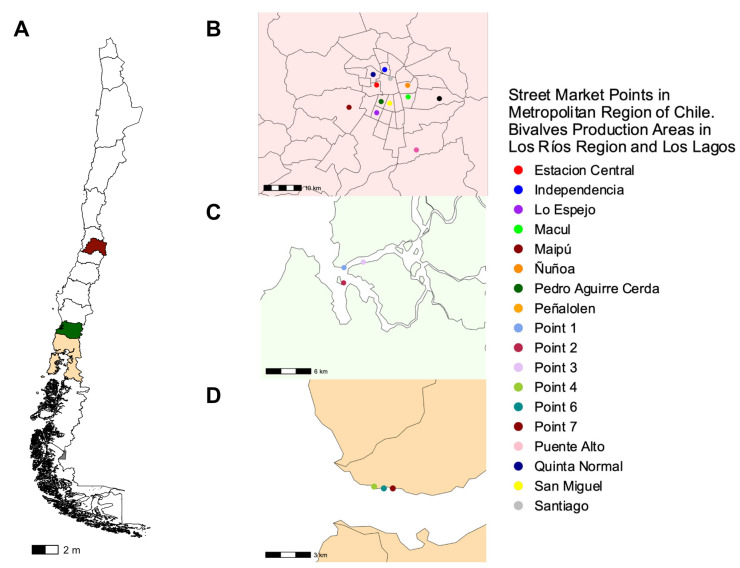
Location map of harvest and street market points in three Regions of Chile. (**A**) The map shows the location of the three Regions of Chile where the sample was collected. (**B**) Additionally, it shows the location of various street markets distributed throughout the city of Santiago in the Metropolitan Region of Chile, which were visited during 2018–2020. (**C**) In addition, it shows the harvest points of the Los Rios Region and (**D**) Los Lagos Region, between 2017 and 2018.

**Figure 4 microorganisms-12-00504-f004:**
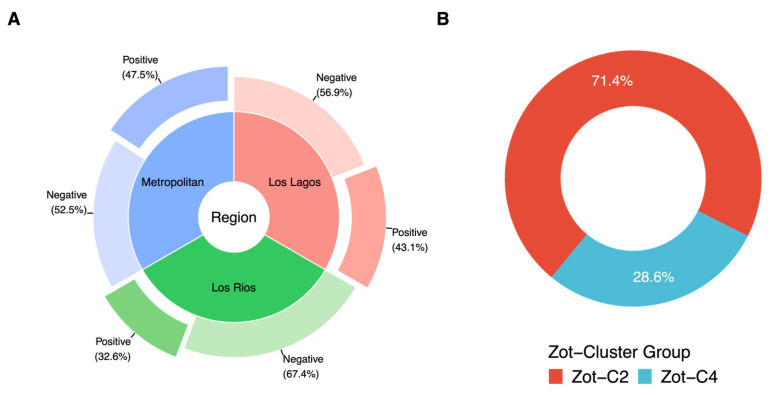
Presence and cluster type group of *zot* gene in *V. parahaemolyticus* isolates obtained from Southern Regions of Chile and street markets in the Metropolitan Region. (**A**) Identification of *zot* gene in 336 isolates of *V. parahaemolyticus* from the Metropolitan Region (blue area), Los Lagos Region (red area) and Los Rios Region (green area). (**B**) Distributions of the 35 *zot* amplicons sequenced in each Zot clusters group. Zot-C2 (C2; red color) and Zot-C4 (C4; sky blue color).

**Figure 5 microorganisms-12-00504-f005:**
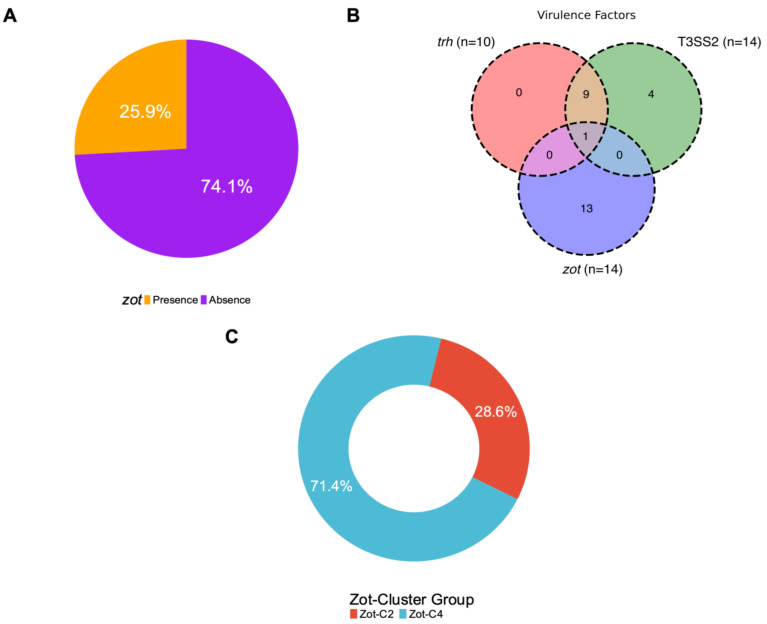
Presence of *zot* and virulence genes in *V. parahaemolyticus* isolates obtained from Northern Adriatic Sea, Italy. (**A**) Percentage distribution of the *zot* gene in 54 isolates. (**B**) Venn diagram of strains distribution with *zot* gene and virulence factors (*trh* gen and T3SS2, α and β phylotype) and co-occurrence of these. (**C**) Classification of *zot* positive amplicons according to the *zot* cluster group. Zot-C2 (C2; red color) and Zot-C4 (C4; sky blue color).

**Figure 6 microorganisms-12-00504-f006:**
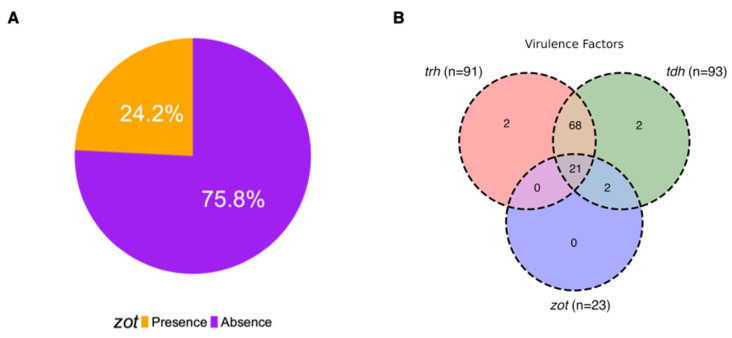
Presence of *zot* and virulence genes in clinical *V. parahaemolyticus* sequences. (**A**) Percentage distribution of the *zot* in clinical isolates. (**B**) Venn diagram of strains distribution with *zot* gene and virulence factors (*tdh* and/or *trh* genes) and the co-occurrence of these.

**Table 1 microorganisms-12-00504-t001:** Zot or *zot*-related sequences described in *V. parahaemolyticus*, based on sequence data available in NCBI.

GeneBank ID	Strain	Year ^1^	Country	Isolation Source	Locus Tag	Description	Ref.
NZ_MKQF01000006.1	PMC53.7	2007	Chile	Clinical	BJL74_RS17965	Zot	[8]
ACFO01000029.1	AN5034	1998	Bangladesh	Clinical	VIPARAN5034_0529	Zot family	[34]
ACKB01000029.1	K5030	2005	India	Clinical	VIPARK5030_1517	Zot family	[34]
ACFM01000008.1	Peru-466	1996	Peru	Clinical	VIPARP466_1523	Zot family	[34]
AFBW01000009.1	10329	1998	USA	Clinical	VP10329_13410	Zot	[35]
CP007004.1	UCM-V493	2002	Spain	Environmental	VPUCM_1658	Zot	[36]
CP009765.1	FORC_006	2014	KOR	Environmental	FORC6_1425	Zot	Unpublished
CP006004.1	01:Kukstr.FDA_R31	2007	USA	Environmental	M634_09525	Toxin	[37]
NZ_CP009847.1	FORC_004	2014	KOR	Environmental	FORC4_1454	Zot	Unpublished
CP009766.1	FORC_006	2014	KOR	Environmental	FORC6_3837	Zot	Unpublished
NZ_CPOO9983.1	FORC_008	2004	KOR	Environmental	FORC8_3762	Zot	[38]
MKQT01000002.1	PMA2.15	2015	Chile	Environmental	BJL84_RS00850	Toxin	[8]
AP000581.2	pO3K6	2016	Japan	Clinical	orf7	Zot like protein	[39]
BA000031.2	RIMD2210633	1996	Japan	Clinical	VP1558	bacteriophage f237 ORF7	[39]
CP013248.1	FORC_022	2015	KOR	Environmental	FORC22_1571	Zot	[40]
CP013827.1	FORC_018	2016	KOR	Environmental	FORC18_3769	Zot	Unpublished
CP011406.1	FORC_014	2015	KOR	Environmental	FORC14_1633	Zot	[41]
CP011407.1	FORC_014	2015	KOR	Environmental	FORC14_3924	Zot	[41]
CP023248.1	MAVP-26	2013	USA	Clinical	YA91_16975	Toxin	[42]
QPIY01000007.1	108B	2018	USA	Environmental	DET53_107186	Zot	Unpublished
CP031781.1	VPD14	2012	China	Environmental	D0853_08290	Hyp protein	Unpublished
CP026041.1	FDAARGOS_51	1998	USA	Clinical	RK51_011060	Toxin	Unpublished

^1^ Correspond to the year of publication of the genome or isolation of the bacterial strain. KOR: South Korea, USA: United States of America. Hyp: Hypothetical.

**Table 2 microorganisms-12-00504-t002:** Presence of *zot* gene in different Chilean environmental and street markets *V. parahaemolyticus* isolates containing *tlh, tdh* and/or *trh* toxins.

		N° of Isolates Containing Toxins					
Region/Year	N° Total Isolates	*tlh*	*tdh*	*trh*	*zot*	*tdh-zot* *	*trh-zot* *
Metropolitan	59	59	0	0	28	0	0
Los Rios	212	212	23	4	69	3	1
Los Lagos	65	65	0	1	28	0	0
**Total**	**336**	**336**	**23**	**5**	**125**	**3**	**1**
2017	173	173	17	5	57	2	1
2018	128	128	6	0	49	1	0
2020	35	35	0	0	19	0	0

* Presence of both genes.

## Data Availability

Data are contained within the article and in the Appendix A.

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
