# Peer review of "Presence of Zonula Occludens Toxin-Coding Genes among Vibrio parahaemolyticus Isolates of Clinical and Environmental Origin"

_microorganisms, 2024, doi:10.3390/microorganisms12030504_

Round 1

Reviewer 1 Report

Comments and Suggestions for Authors

In this paper, authors accurately analyzed the presence of the zonula occludens toxin (zot) gene, which encodes the homonymous toxin causing cytoskeletal damage in Caco-2 infected cells, in non-toxigenic V. parahaemolyticus strain PMC53.7 environmental and clinical isolates coming from Chile and Italy. To do so, universal zot primers were designed. The results revealed the presence of this gene both in Chilean and Italian samples, in fact around 37.2% of Chilean environmental isolates and 25.9% of Italian ones carried the zot gene, so as around the 24.2% of the clinical isolates, showing the importance of further studies aimed to understand the role of the ZOT in V. parahaemolyticus virulence.  

However, there are several issues that need to be addressed before publication:

- line 1: italic in “zonula occludens toxing-coding genes” is not required as you are referring to the toxin and not to the gene. If you want to refer to the gene, please maintain the italic formatting but remove the word “-coding”.

- line 24: italics is not required as you are referring to the toxin and not to the gene and please format “zot” in uppercase.

- line 30: please replace “Italia” with “Italy”.

- line 48: please replace “for colonize” with “for colonizing”.

- line 59: italics is not required as you are referring to the toxin and not to the gene and please format “zot” in uppercase.

- lines 77 and 79: italics in “zot” is not required as you are referring to the toxin and not to the gene. If you want to maintain the italics formatting, please remove the word “-coding”.

- line 78: please replace “to delucidated” with “the elucidate” or “to clarify”.

- line 83: please replace “Santiago of Metropolitan Region” with “Santiago metropolitan Region”.

- line 89: please replace “In Metropolitan region” with “IN the Metropolitan Region”.

- line 120: please replace “Primer’s design” with “Primers design”.

- line 151: please replace "contain” with “contains”.

- line 166: please get the table caption underneath so that it is placed right above table 1.

- line 183: please remove “but”.

- line 192: please replace “the optimal primer pair designed as” with “the optimal primer pair were designed as”.

- line 215: please replace “Primer’s” with “Primers’ ”.

- lines 216-219: please revise the whole sentence

- line 220: please replace “cycle” with “cycles”.

- line 223: according to fig. 2, line 1 is used as a negative control without gDNA. Maybe you should  replace “Lane 1” with “Lane 4”?

- line 229: please replace “it was confirmed PMC53.7” with “it was confirmed that PMC53.7”.

- line 242: please replace “59 strain” with “59 strains” and “Metropolitana” with “Metropolitan”.

- line 243: please remove the comma after “Region” and replace “encompass” with “and encompassed” or “encompassing”.

- line 246: please replace “212 strain” with “212 strains”.

- line 254: please replace “detect” with “detected”.

- line 256: please replace “gen” with “gene”.

- line 278: please replace “only detected” “it was only detected”. 

- line 288: please replace “and the” with “and if the”.

- line 295: please replace “showed a” with” showing the”.

- line 296: please replace “showing a” with “pointing out the”. 

- line 376: please replace “were” with “was”.

- line 378: please remove the comma after the word “study” and replace “relevance” with “relevant”.

- lines 403-405: please replace with “Our study revealed the presence of zot in V. parahaemolyticus isolates coming from Los Rios Region (32.5%), Los Lagos Region (43.1%), and the street market of the Metropolitan Region (47.5%).”

- References: please revise the style according to the journal’s standards

Comments on the Quality of English Language

Only some minor editing and/or some typos

Author Response

Dear reviewer,

We fully appreciate your time and dedication in reviewing our article. Below we will resolve your concerns:

line 1: Thanks for your comment. We have written in italics considering it as a Latin expressions and following RAE. But we are agreed with you, and we will remove the italics.
line 24: Changed without italic letters
line 30: replaced as recommended.
line 48: replaced as recommended.
line 59: replaced as recommended.
lines 77 and 79: Changed without italic letters
line 78: replaced as recommended.
line 83: replaced as recommended.
line 89: replaced as recommended.
line 120: replaced as recommended.
line 151: replaced as recommended.
line 166: replaced as recommended, it was an error in tempering.
line 183: change made.
line 192: replaced as recommended.
line 215: replaced as recommended.
lines 216-219: the sentence was corrected and changed to improve its readability and explanation according to the recommendations
line 220: replaced as recommended.
line 223: replaced according to recommendation, it was an error on our part when indicating the lane.
line 229: replaced as recommended.
line 242: replaced as recommended.
line 243: replaced as recommended.
  line 246: replaced according to recommendation.
line 254: replaced as recommended.
  line 256: replaced as recommended.
line 278: replaced as recommended.
line 288: replaced as recommended.
line 295: replaced as recommended.
line 296: replaced as recommended.
  line 376: replaced as recommended.
line 378: replaced as recommended.
lines 403-405: replaced as recommended, and we appreciate your suggestion to help understand the manuscript.
References: Changes were made to the necessary italics and capitalization.

We really appreciate all your suggestions and recommendations for improving the manuscript, which were fully considered for the latest attached version.

Greetings and thank you very much.

Reviewer 2 Report

Comments and Suggestions for Authors

Short Brief summary

The authors collected 336 environmental isolates of Vibrio parahaemolyticus from Chile (336 strains), 95 clinical isolates (95 strains) and 54 environmental isolates from Italy (54 strains) and examined the distribution of zot genes, which are important virulence factors. 22 zot genes were clustered into 4 groups, zot-C1~4, based on their sequences. In this study, we examined the presence or absence of zot genes, their clusters if present, and their co-localization with other virulence genes in these strains.

General concept comments

L72

Based on this evidence, ...

The preceding sentence and the references discuss the findings on Vibrio cholerae. To return to the topic of Vibrio parahaemolyticus, it would be better to mention how references 18 and 19 link to Vibrio parahaemolyticus.

L74 other virulence factors

Name again tdh, tlh and trh for specific genes.

L268Fig4A

The text in the pie chart is crushed and difficult to read. Make an effort to increase the resolution or rework the labels.

L259

It is unfortunate that you did not cluster all the isolates. I assume that it was probably not possible due to the amount of work involved. It should be noted that you did not cluster all isolates in the discussion.

L309 tdh and/or trh genes

There is no letter tdh in the figure, but T3SS2. Is this correct?

Fig4B and fig5C should be in the same color scheme as they are, but Fig5A and Fig6A should be in the same style, but in a different color scheme than Fig4B.

Author Response

Dear reviewer,

We fully appreciate your time and dedication in reviewing our article. Below we will resolve your concerns:
L72
Based on this evidence,...
The preceding sentence and the references discuss the findings on Vibrio cholerae. To return to the topic of Vibrio parahaemolyticus, it would be better to mention how references 18 and 19 link to Vibrio parahaemolyticus.

- A paragraph was added connecting the ideas, especially mentioning what was stated in references 18 and 19. "While there are currently no reported associations of the zot gene with diarrhea outbreaks caused by V. parahaemolyticus, the absence of such associations may explain the lack of prevalence studies on zot in this species. However, the presence of zot in non-toxigenic strains of V. cholerae raises intriguing questions about its potential role in other bacterial species. Specifically, it raises the necessity to investigate whether zot might be associated with V. parahaemolyticus strains lacking classical virulence markers such as hemolysins or T3SS2, especially clinical isolates."

L74 other virulence factors
Name again tdh, tlh and trh for specific genes.

- Added what was recommended by the reviewer, thank you very much.

L268Fig4A
The text in the foot chart is crushed and difficult to read. Make an effort to increase the resolution or rework the labels.

- The quality of the aforementioned figure and others was improved for better understanding-

L259
It is unfortunate that you did not cluster all the isolates. I assume that it was probably not possible due to the amount of work involved. It should be noted that you did not cluster all isolates in the discussion.

-It was decided not to group the strains in their entirety so as not to confuse the reader with the percentages or to make it easier for readers to find environmental or clinical information, so only some cases were highlighted in the conclusion. We hope you can understand the case-

L309 tdh and/or trh genes
There is no letter tdh in the figure, but T3SS2. Is this correct?
Fig4B and fig5C should be in the same color scheme as they are, but Fig5A and Fig6A should be in the same style, but in a different color scheme than Fig4B.

- It is correct, but at the same time, thanks to your comment, the figure caption was improved for greater understanding, highlighting that they do not have TDH but do have T3SS2 (alpha or beta type).
In addition, the colors of the schemes were changed as recommended in the aforementioned figures, considering this change important for understanding and greatly thanking the reviewer.